# Modulation of formin processivity by profilin and mechanical tension

Luyan Cao[1†], Mikael Kerleau[1†], Emiko L. Suzuki[1], Hugo Wioland[1], Sandy Jouet[1], Berengere Guichard[1], Martin Lenz[2], Guillaume Romet-Lemonne[1]*, Antoine Jegou[1]*

[1]Institut Jacques Monod, CNRS, Université Paris Diderot, Paris, France; [2]LPTMS, CNRS, Université Paris-Sud, Université Paris-Saclay, Orsay, France

**Abstract** Formins are major regulators of actin networks. They enhance actin filament dynamics by remaining processively bound to filament barbed ends. How biochemical and mechanical factors affect formin processivity are open questions. Monitoring individual actin filaments in a microfluidic flow, we report that formins mDia1 and mDia2 dissociate faster under higher ionic strength and when actin concentration is increased. Profilin, known to increase the elongation rate of formin-associated filaments, surprisingly decreases the formin dissociation rate, by bringing formin FH1 domains in transient contact with the barbed end. In contrast, piconewton tensile forces applied to actin filaments accelerate formin dissociation by orders of magnitude, largely overcoming profilin-mediated stabilization. We developed a model of formin conformations showing that our data indicates the existence of two different dissociation pathways, with force favoring one over the other. How cells limit formin dissociation under tension is now a key question for future studies.
DOI: https://doi.org/10.7554/eLife.34176.001

*For correspondence:
guillaume.romet-lemonne@ijm.fr (GR-L);
antoine.jegou@ijm.fr (AJ)

[†]These authors contributed equally to this work

Competing interests: The authors declare that no competing interests exist.

## Introduction

The diversity of actin filament networks in cells stems from a few key nucleators, such as formins and the Arp2/3 complex, which have very specific activities (*Blanchoin et al., 2014*; *Bovellan et al., 2014*; *Wales et al., 2016*). In cells, formins are responsible for the generation of elongated, unbranched actin filament structures such as the ones found in filopodia, stress fibers, the cytokinetic ring, and within the nucleus (*Isogai and Innocenti, 2016*). Formin malfunction is linked to a number of pathologies, such as angiogenesis (*Phng et al., 2015*), neuropathies (*Roos et al., 2015*) and cancer (*Choi et al., 2016*).

Formins function as homodimers and most isoforms share a similar mode of activation, where the interaction of activators with N-terminal domains releases auto-inhibition and mediates the anchoring of formins to membranes. Formin functional domains, Formin Homology Domains 1 (FH1) and 2 (FH2), are responsible for their most salient features: their ability to track both growing and depolymerizing filament barbed ends and to accelerate their elongation from profilin-actin (*Higashida et al., 2004*; *Jégou et al., 2013*; *Kovar and Pollard, 2004*; *Mizuno et al., 2011*; *Romero et al., 2004*). Rapid elongation is achieved by the FH1 domains, seen as flexible chains containing polyproline tracks, which bind profilin-actin complexes and deliver them to the barbed end (*Higashida et al., 2004*; *Kovar and Pollard, 2004*; *Romero et al., 2004*). Barbed end tracking is achieved by the translocation of the FH2 dimer, which encircles the actin subunits at the barbed end (*Otomo et al., 2005*).

Formin processivity, quantified by the dissociation rate of the formin from the barbed end, determines for how long filaments interact with a formin. While a formin resides at the barbed end, it decreases barbed end affinity for Capping Protein (*Bombardier et al., 2015*; *Shekhar et al., 2015*; *Zigmond et al., 2003*), modulates its elongation, and can maintain it anchored to a membrane.

Processivity is thus a pivotal characteristic, determining formins' ability to shape filament networks and transmit forces.

Formin processivity has long been identified as an essential feature of formins and occasional measurements have revealed quantitative differences between isoforms (*Bilancia et al., 2014*; *Kovar et al., 2006*; *Paul and Pollard, 2008*; *Romero et al., 2004*; *Vizcarra et al., 2014*). Negative regulators bind to FH2 to displace formin from filament barbed ends (*Chesarone et al., 2009*; *Chesarone-Cataldo et al., 2011*), whereas Ena/VASP, via its EVH1 domain, is able to bind to FH1 domains without impacting formin processivity (*Bilancia et al., 2014*). While processivity seems mainly governed by FH2-actin interactions, the DAD domain (or 'tail'), found next to the FH2 domain at the C-terminus, has been reported to contribute to the processivity of Drosophila formin Capuccino (*Vizcarra et al., 2014*). The dissociation rate of yeast formin Bni1p has been proposed to scale with filament elongation velocity, suggesting the existence of a transient, weakly bound state occurring upon actin subunit addition (*Paul and Pollard, 2008*).

Today, many important aspects of formin processivity remain unclear. The possible involvement of formin's other domains and the modulation of formin processivity by various physiological factors have yet to be determined. In particular, pulling forces such as the ones exerted on actin filaments in cells (*Romet-Lemonne and Jégou, 2013*) have been reported to modulate formin elongation (*Courtemanche et al., 2013*; *Jégou et al., 2013*; *Kubota et al., 2017*; *Yu et al., 2017*; *Zimmermann et al., 2017*) but their impact on processivity is an open question.

Here, we systematically quantify the dissociation rate of mammalian formins mDia1 and mDia2 in different in vitro conditions. Using microfluidics to monitor and manipulate individual actin filaments (*Figure 1*), we find that the dissociation rate is modulated by ionic strength (*Figure 2*) as well as by actin and profilin concentrations (*Figure 3*). Profilin prolongs formin residence at the barbed end via its interaction with the FH1 domain, allowing rapid elongations without enhancing formin dissociation. We find that tension applied on filaments has a dramatic impact on the formin dissociation rate, which increases by several orders of magnitude, independently of other parameters (*Figure 4*). We compare the impact of these different factors on the typical lengths reached by the formin-elongated filaments (*Figure 5*). A mathematical model describing the possible formin states at the barbed end is developed and confronted to our experimental data (*Figure 6*). It indicates that, when an actin subunit is added to the barbed end, the formin goes through a dissociation-prone transition, which is relatively insensitive to force, and which can be stabilized by FH1-profilin-barbed end interactions.

## Results

### Single-filament Microfluidics is an efficient means to measure formin processivity under various conditions

We have carried out experiments using a standard microfluidics chamber with three inlets, in different configurations (*Figure 1A–C* and Materials and methods). Using anchored spectrin-actin seeds we have monitored the growth of free actin filaments barbed ends, which we exposed to a solution of formin for typically ten seconds and resumed exposing to constant concentrations of actin and profilin. The presence of formin at the barbed end was visible thanks to its faster elongation (in the presence of profilin). This configuration was used with fluorescently labeled actin (*Figure 1A*) or alternating exposure to labeled actin with unlabeled actin, producing striped filaments, which allowed us to measure formin related rate constants when incorporating fully unlabeled actin segments (*Figure 1B*, Materials and methods). Another configuration consisted in anchoring formins to the coverslip surface, nucleating and elongating filaments from these formins (*Figure 1C*). This allowed us to monitor the elongation of filaments from unlabeled actin, and the dissociation of the formin from the barbed end was revealed by the detachment of the filament which is then carried away by the flow. This configuration applies calibrated forces to the filament-formin interaction (*Jégou et al., 2013*), which can be kept very low (<0.1 pN) using a low microfluidics flow rate, or made significant, up to several pN, by increasing the flow rate (*Figure 4*). Except for the variant with striped filaments, we have used these experimental configurations in earlier studies (*Jégou et al., 2013*; *Montaville et al., 2014*; *Shekhar et al., 2015*).

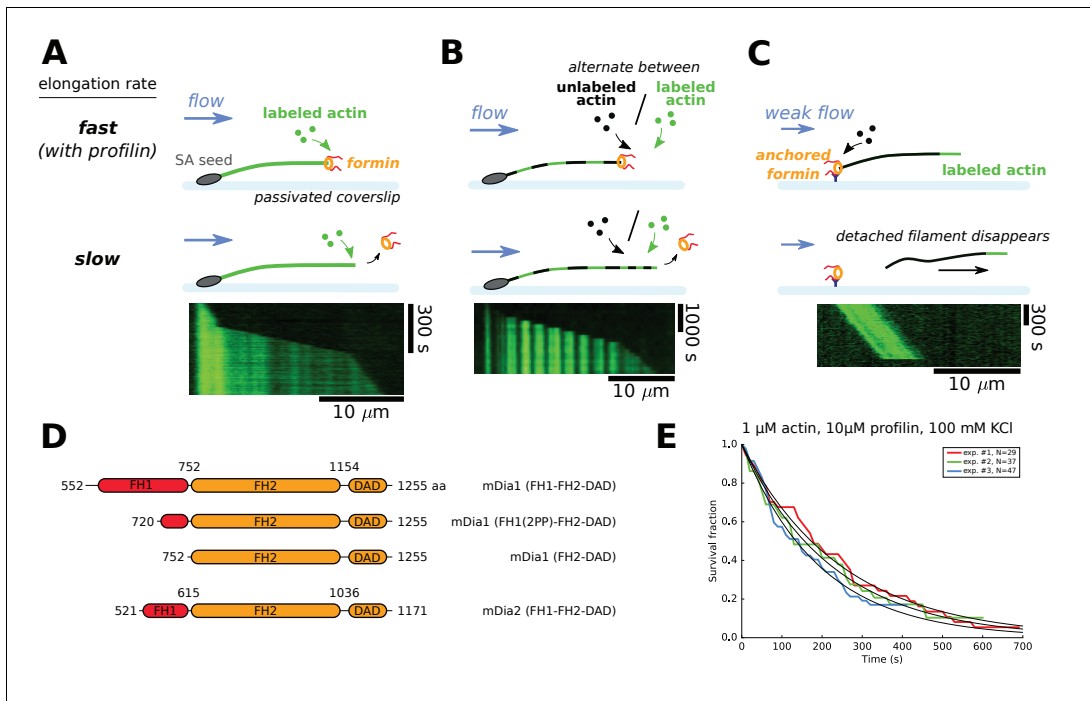

**Figure 1.** Single-filament microfluidics experimental configurations to measure formin processivity. (**A–C**) Different experimental configurations using microfluidics for the study of formin processivity, showing sketches of the side view (top) and typical kymographs of individual filaments (bottom). (**A**) Alexa 488 labeled actin filaments are elongated from surface-anchored spectrin-actin seeds. Transient exposure to a formin solution puts formins on filament barbed ends, which elongate faster (here in the presence of 1 µM 15% Alexa 488 labeled actin +5 µM profilin, at 100 mM KCl). Upon formin dissociation, the barbed end elongates slower. Images were acquired in TIRF microscopy. See **Video 1**. (**B**) Same configuration as in (**A**), but the filaments are exposed to a periodic alternation of different conditions: here a solution of unlabeled actin (0.3 µM actin, 50 mM KCl) for 100 s and a solution of 15% Alexa 488 labeled actin (0.5 µM actin +2 µM profilin, 50 mM KCl) for 20 s. Images were acquired in epifluorescence while exposing to unlabeled actin. See **Video 2**. (**C**) Configuration where formins are anchored to the surface by their C-terminus. Filaments were nucleated using a solution of labeled actin and elongated by flowing in a solution of unlabeled actin (here, 0.3 µM actin, at 50 mM KCl), until the filaments eventually detached and disappeared. The viscous drag applied on the filaments was kept low (<0.1 pN) by working with low flow rates. Images were acquired in epifluorescence. See **Video 3**. (**D**) Domain architecture and boundaries for the mDia1 and mDia2 formin constructs used in this study. (**E**) Survival fractions of mDia1(FH1-FH2-DAD) formin-bound barbed ends as a function of time, obtained from three independent experiments performed in the same conditions, in the experimental configuration shown in (**A**). Curves are fitted by a mono-exponential decay to obtain formin dissociation rate $k_{off}$.

DOI: https://doi.org/10.7554/eLife.34176.002

These different configurations allowed us to measure, under a given set of conditions, the survival fraction of filaments that still bear a formin at their barbed end as a function of time (**Figure 1E**), giving access to the formin dissociation rate constant $k_{off}$. The experimental configurations shown in **Figure 1B and C** were specifically used to determine $k_{off}$ with unlabeled actin or with no profilin. We have verified that the results were not affected by our choice of experimental configuration.

We used purified actin from rabbit muscle, either unlabeled or labeled on lysine 328 with Alexa 488 (**Tóth et al., 2016**). We used recombinant formin constructs (**Figure 1D**): mDia1(FH1-FH2-DAD) with full length functional domains; a truncated mDia1(FH1(2PP)-FH2-DAD) with an FH1 domain that contained only the two polyproline (PP) tracks closest to the FH2 domain; mDia1(FH2-DAD) which contained no FH1 domain at all; and mDia2(FH1-FH2-DAD).

## Impact of ionic strength and actin labeling on formin processivity

Varying KCl concentration in our assay buffer (see Materials and methods), we found that the ionic strength had a strong impact on formin dissociation (**Figure 2A,B**). In comparison, the same

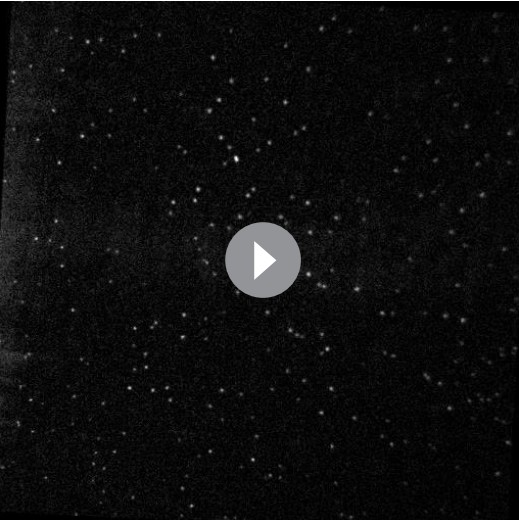

**Video 1.** Actin filaments elongating with 1 μM 15% Alexa 488-labeled actin and 5 μM profilin, at 100 mM KCl, are transiently exposed to a solution of 20 nM mDia1(FH1-FH2-DAD) for 20 s (during frames 27–30). Images were acquired in TIRF. Full field of view is 137 × 137 μm. Interval between images is 5 s (movie is accelerated 75x). The solution flows from left to right. Corresponds to *Figure 1A* of the main text.
DOI: https://doi.org/10.7554/eLife.34176.003

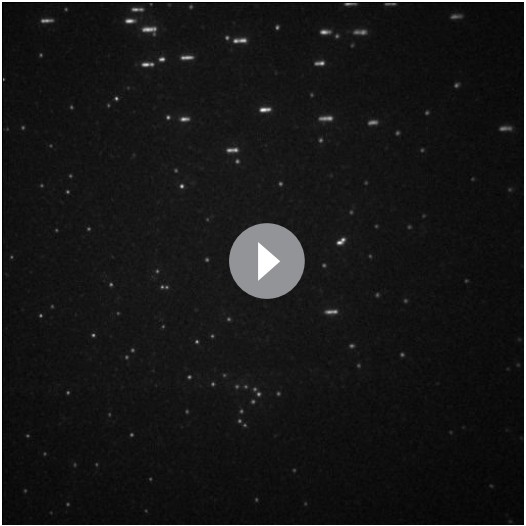

**Video 2.** Actin filaments are exposed to a periodic alternation of a solution of unlabeled actin (0.3 μM actin, 50 mM KCl) for 100 s and a solution of 15% Alexa 488-labeled actin (0.5 μM actin +2 μM profilin, 50 mM KCl) for 20 s. The filaments are transiently exposed to a solution of 11 nM mDia1(FH1-FH2-DAD) for 5 s, after frame number 5. Images were acquired in epifluorescence while exposing to unlabeled actin. Full field of view is 137 × 137 μm. Interval between images is 120 s (movie is accelerated 360x). The solution flows from left to right. Corresponds to *Figure 1B* of the main text.
DOI: https://doi.org/10.7554/eLife.34176.004

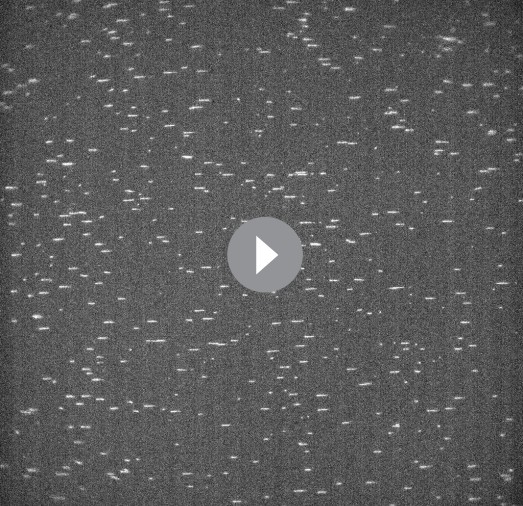

**Video 3.** Actin filaments were nucleated from surface-anchored formins mDia1(FH1-FH2-DAD) with 15% Alexa 488-labeled actin, and elongate with 0.3 μM unlabeled actin, at 50 mM KCl. Full field of view is 221 × 221 μm. Images were acquired in epifluorescence. Interval between images is 10 s (movie is accelerated 70x). A minimal flow is applied. The solution flows from left to right. Corresponds to *Figure 1C* of the main text.
DOI: https://doi.org/10.7554/eLife.34176.005

variations of the ionic strength had a limited impact on the barbed end elongation rate, with or without formins (*Figure 2B*, inset). In order for formin dissociation rates to be in a range that could be measured accurately, we have used either 50 or 100 mM KCl depending on whether we were studying mDia1 or mDia2 formins, and whether mechanical tension was applied. We have verified that the effects we report in the rest of this paper are not qualitatively affected by the choice of ionic strength (*Figure 2—figure supplement 1*).

Labeling actin with a fluorophore can hinder its polymerization or its interaction with regulatory proteins (*Chen et al., 2012*; *Kuhn and Pollard, 2005*) and lead to unsuspected artefacts (*Niedermayer et al., 2012*). Here, our labeling of actin on lysine 328 with Alexa 488 fluorophore had no measurable impact on the elongation rate of formin-free barbed ends, but slowed down their elongation with formins significantly and enhanced formin dissociation rate (*Figure 2C,D*). Using our microfluidics setup to measure reaction rates with unlabeled actin (*Figure 1B,C*), we have

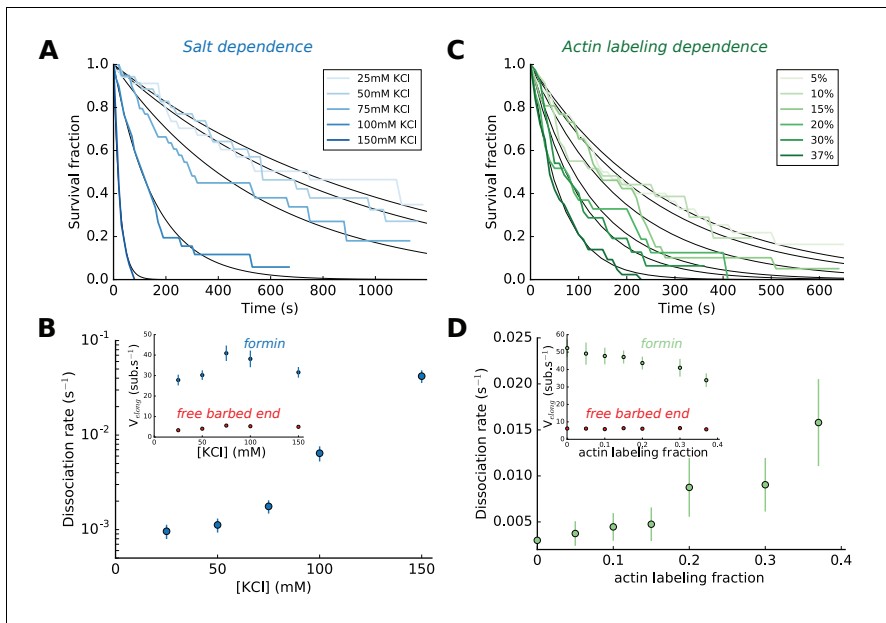

**Figure 2.** Impact of salt and actin labeling fraction on mDia1 formin processivity. (**A,B**) Effect of salt concentration on the survival fraction of formin-bound barbed ends (**A**), on the formin dissociation rates (**B**, log-linear scale) as well as on the barbed end elongation rates (**B**, inset). The dissociation rates in (**B**) result from the exponential fits (black lines) shown in (**A**). Each data point corresponds to a population of 30–40 filaments. (**C,D**) Effect of the actin Alexa 488 labeling fraction on the survival fraction of formin-bound barbed ends (**C**), on the formin dissociation rates (**D**) and on the barbed end elongation rates (**D**, inset). Each data point in (**D**) corresponds to a population of 30–40 filaments. Error bars on formin dissociation rates indicate 65% confidence intervals based on exponential fits and sample size (see Materials and methods), and error bars on elongation rates indicate standard deviations.

DOI: https://doi.org/10.7554/eLife.34176.006

The following source data and figure supplement are available for figure 2:

**Source data 1.** Spreadsheet containing the data plotted in *Figure 2* and *Figure 2—figure supplement 1*.
DOI: https://doi.org/10.7554/eLife.34176.008

**Figure supplement 1.** Variation of the formin mDia1(FH1-FH2-DAD) dissociation rate as a function of profilin concentration, for 2 μM unlabeled actin at 50 mM KCl (red) or for 1 μM unlabeled actin at 100 mM KCl (blue), showing that formin processivity is decreased by profilin with unlabeled actin, for both salt conditions.
DOI: https://doi.org/10.7554/eLife.34176.007

verified that the conclusions we drew from the observation of 15% Alexa 488-labeled actin filaments were not biased by labeling (*Figure 2—figure supplement 1*).

## Profilin increases formin processivity, involving FH1 domains

For a given profilin concentration, the barbed end elongation rate $v_{elong}$ scales with the actin concentration (with or without formin, *Figure 3—figure supplement 1*) and we observed that the formin dissociation rate $k_{off}$ increased with actin concentration, and thus with the elongation rate (*Figure 3A,C*). This confirmed earlier observations on yeast formin Bni1p (*Paul and Pollard, 2008*). However, for both mDia1 and mDia2, the amplitude of the increase of the formin dissociation rate with actin concentration appeared to depend significantly on profilin concentration (*Figure 3A,C*). As a result, there is no universal scaling of $k_{off}$ with the elongation rate. In fact, using different sets of actin and profilin concentrations, one can obtain identical elongation rates with very different formin dissociation rates.

To investigate this point further, we measured the formin dissociation rate as a function of profilin concentration at a fixed actin concentration and found that $k_{off}$ decreased with increasing profilin concentration (*Figure 3B,D*). In contrast with actin, the modulation of the elongation rate by profilin is biphasic (*Kovar et al., 2006*): low profilin concentrations increase $v_{elong}$ as actin becomes profilin-actin, while higher concentrations slow down elongation as excess profilin competes with profilin-actin for polyproline binding sites on FH1 domains and barbed ends (*Figure 3B,D* insets).

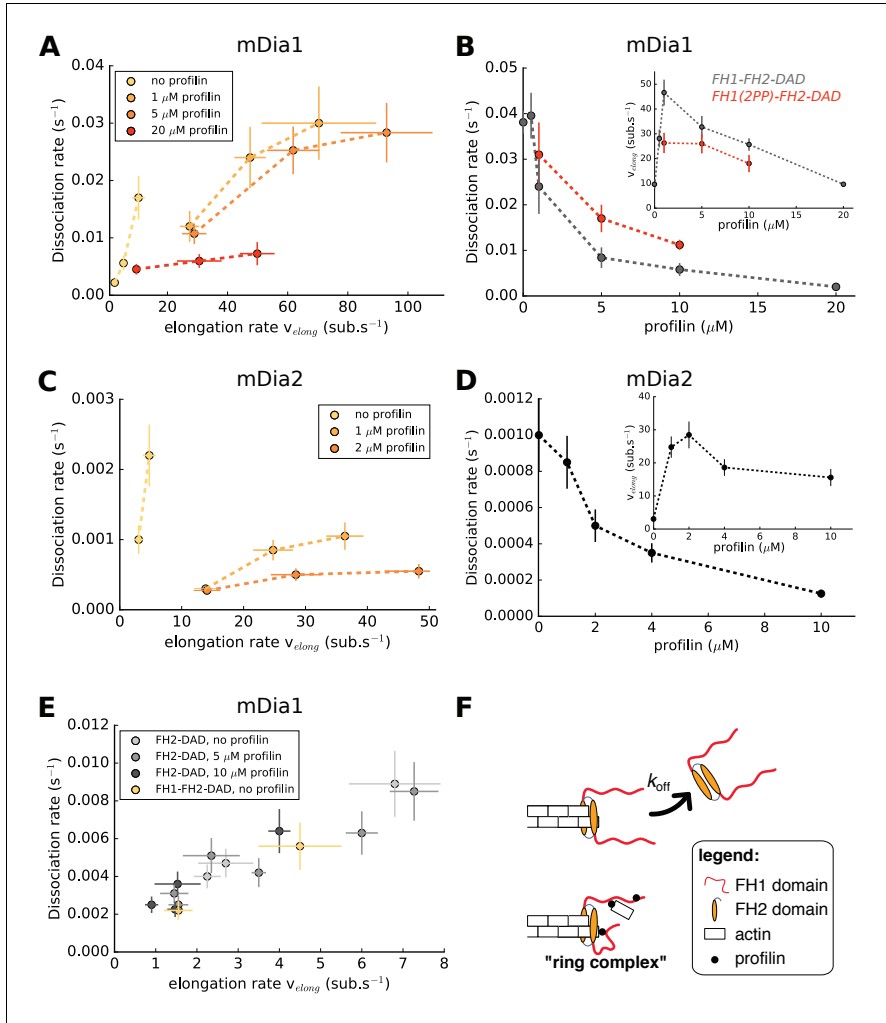

**Figure 3.** Formin dissociation is enhanced by G-actin concentration, and slowed down by profilin. (A, C) Variation of the mDia1 (A) or mDia2 (C) formin dissociation rate as a function of the barbed end elongation rate. Each data set (N = 30–40 filaments) was obtained with a fixed profilin concentration and different actin concentrations, at 100 mM KCl. Each point corresponds to an independent experiment, performed with 15% Alexa 488-labeled actin, except for the data of mDia1 without profilin which were acquired with unlabeled actin (as described in *Figure 1B*). (B, D) Variation of the formin dissociation rate and the barbed end elongation rate (inset) as a function of profilin concentration, (B) for mDia1 formins with a full length FH1 (FH1-FH2-DAD) and with a truncated FH1 containing only two polyproline tracks (FH1(2PP)-FH2-DAD), and (D) for mDia2(FH1-FH2-DAD). The data was obtained with 15% Alexa 488-labeled actin, at 100 mM KCl. The same profilin dependence was observed using unlabeled actin, for both 50 and 100 mM KCl (*Figure 2—figure supplement 1*). Each data point corresponds to the average of 1–3 independent experiments. (E) Variation of the formin dissociation rate as a function of the barbed end elongation rate: for mDia1(FH2-DAD) homodimers in the presence or absence of profilin, and for mDia1(FH1-FH2-DAD) in the absence of profilin, all with unlabeled actin. (F) Sketch illustrating the profilin-mediated interaction between FH1 and the barbed end, forming the 'ring complex', which appears to prevent the dissociation of formin from the barbed end. Error bars on formin dissociation rates indicate 65% confidence intervals based on exponential fits and sample size (see Materials and methods), and error bars on elongation rates indicate standard deviations.

DOI: https://doi.org/10.7554/eLife.34176.009

The following source data and figure supplements are available for figure 3:

**Source data 1.** Spreadsheet containing the data plotted in *Figure 3*, *Figure 3—figure supplement 1*, and *Figure 3—figure supplement 2*.

DOI: https://doi.org/10.7554/eLife.34176.012

*Figure 3 continued*

**Figure supplement 1.** Variation of the mDia1 formin-bound or free barbed end elongation rate $v_{elong}$ as a function of actin concentration.

DOI: https://doi.org/10.7554/eLife.34176.010

**Figure supplement 2.** Mutant profilin-R88E has no impact on the processivity of mDia1.

DOI: https://doi.org/10.7554/eLife.34176.011

Importantly, the decrease of $k_{off}$ was also observed in the lower range of profilin concentrations, where the elongation rate greatly increases with profilin. It thus appears that profilin itself reduces formin detachment, independently of the barbed end elongation rate.

In order to estimate the role of the FH1 domains in the profilin-induced reduction of the dissociation rate, we repeated these measurements using a truncated mDia1 formin construct, FH1(2PP)-FH2-DAD, where both FH1 domains of the formin homodimer only contained two profilin-binding polyproline tracks. We found that the truncated formin still enhanced filament elongation from profilin-actin, though not as strongly as the formin with full-length FH1 (*Figure 3B* inset). It still exhibited a reduction of $k_{off}$ with profilin concentration (*Figure 3B*), but the dissociation rate of FH1(2PP)-FH2-DAD was consistently higher than of wild type FH1-FH2-DAD for all profilin concentrations tested. These results confirm that the formin dissociation rate does not generally scale with the elongation rate. They also suggest that FH1 polyproline tracks, which are responsible for rapid elongation, are also responsible for the decrease of $k_{off}$ in the presence of profilin.

To further investigate the contribution of the FH1 domains, we then asked whether the reduction of the dissociation rate by profilin required its binding to the FH1 domain, or if the rapid equilibrium of profilin with the barbed end was enough to stabilize its interaction with the formin. We reasoned that if the latter hypothesis was correct, the processivity of mDia1(FH2-DAD) dimers (with no FH1 domains) should be enhanced by the binding of profilin to the barbed end. To test this, we compared the FH2 dimer dissociation rate for different barbed end elongation rates, obtained in the presence or absence of profilin (*Figure 3E*). We found that the presence of a large excess of profilin, which significantly puts the barbed end in a profilin-bound state and slows down its elongation (*Jégou et al., 2011*; *Pernier et al., 2016*), led to the same FH2 dimer dissociation rate as when the same elongation rates were reached without profilin. Thus, in the absence of FH1 domains, profilin simply reduces the dissociation rate of the FH2 dimer by slowing down barbed elongation. These results indicate that FH1 is required in order for profilin to decrease the formin dissociation rate $k_{off}$ independently of the barbed end elongation rate. In the absence of profilin, FH1-FH2 behaved like FH2 (*Figure 3E*), indicating that the presence of FH1 domains alone, in the absence of profilin, has no impact on processivity. Mutant profilin-R88E, which binds to FH1 but not to actin (*Kovar et al., 2006*; *Lu and Pollard, 2001*), has no impact on processivity (*Figure 3—figure supplement 2*), indicating that the interaction of profilin with G-actin and/or filament barbed ends is required in order to decrease the formin dissociation rate.

## Mechanical tension strongly decreases formin processivity

In cells, anchored formins are exposed to mechanical tension applied to actin filaments, typically as a consequence of myosin activity. We thus investigated the impact of such forces on formin processivity. To do so, we performed experiments with surface-anchored formins, in the configuration shown in *Figure 1C*, but using higher flow rates in order to apply significant tension to the filaments (*Figure 4A*). In a previous study, we have shown that the force at the anchoring point scales with the filament length (*Jégou et al., 2013*), and thus increases as the filaments elongate over time. Here, the sigmoidal shape of the survival fractions over time indicated an increase of the dissociation rate $k_{off}$ with the applied force (*Figure 4B*). In order to avoid making assumptions regarding the force-dependence of the dissociation rate $k_{off}$, we determined $k_{off}$ at different forces by local fits of the survival fractions (see Materials and methods). We verified that the filament detachment events observed during the experiment corresponded to filament-formin dissociations (as sketched in *Figure 4A*) by checking that formins were still on the surface at the end of the experiment (see Materials and methods and *Figure 4—figure supplement 1*).

We found that mechanical tension had a dramatic impact on the formin dissociation rate, which increased by a few orders of magnitude when piconewton forces were applied (*Figure 4C–F*).

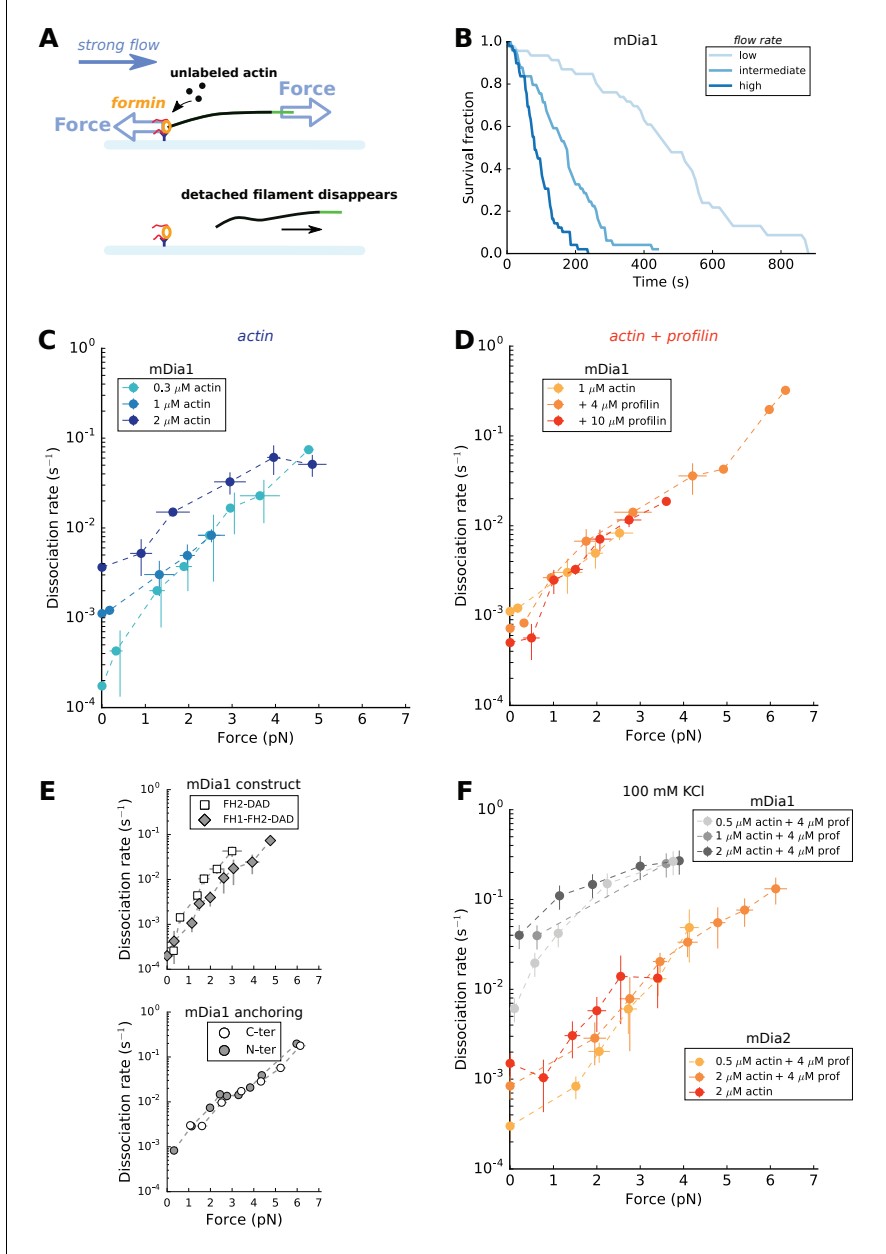

**Figure 4.** Force has a great impact on formin processivity. (**A**) Sketch of the experimental configuration, similar to that of *Figure 1C*, but where significant forces are applied using various flow rates. The applied force scales with the filament length. (**B**) Survival fractions of mDia1-anchored filaments, elongating with 1 µM actin +10 µM profilin, using different flow rates to reach different force ranges: each filament underwent 0.051 pN/µm (initial filament length = 4.9 µm, N = 46 filaments), 0.204 pN/µm (initial filament length = 3.2 µm, N = 49) or 0.501 pN/µm (initial filament length = 2.6 µm, N = 49) for 'low', 'medium' and 'high' flow rate curves, respectively. (**C–E**) mDia1 formin dissociation rate as a function of applied force (log-linear plots), for different actin concentrations in the absence of profilin (**C**); for 1 µM actin with different profilin concentrations (**D**); for 0.3 µM actin in presence or absence of FH1 domains (E, top); and for 1 µM actin, 4 µM profilin for mDia1 (FH1-FH2-DAD) formins either anchored by their FH1 N-terminus or FH2 C-terminus (E, bottom). Experiments were carried out by elongating the filaments with unlabeled actin, at 50 mM KCl. (**F**) mDia1 and mDia2 formin dissociation rates as a function of applied force (log-linear plots), for different profilin and unlabeled actin concentrations, at 100 mM KCl. Dissociation rates were obtained by local fits of the slope in survival fractions similar to the ones shown in (**B**) (see Materials and methods). Each data point is either obtained from a single experiment or is the average of 2–3 independent experiments. The data points at zero force were measured independently, using the configuration shown in *Figure 1B* (striped

*Figure 4 continued on next page*

*Figure 4 continued*

filaments). The error bars indicate standard deviations when several independent experiments were grouped (data from individual experiments for (**C**) and (**D**) are shown in Supp. *Figure 4—figure supplement 2*).

DOI: https://doi.org/10.7554/eLife.34176.013

The following source data and figure supplements are available for figure 4:

**Source data 1.** Spreadsheet containing the data plotted in *Figure 4*, *Figure 4—figure supplement 1*, *Figure 4—figure supplement 2*, and *Figure 4—figure supplement 3*.

DOI: https://doi.org/10.7554/eLife.34176.017

**Figure supplement 1.** C-terminus anchored mDia1 (FH1-FH2-DAD) formin renucleation.

DOI: https://doi.org/10.7554/eLife.34176.014

**Figure supplement 2.** mDia1 formin dissociates faster with force.

DOI: https://doi.org/10.7554/eLife.34176.015

**Figure supplement 3.** mDia2 formin elongation rate is mechanosensitive.

DOI: https://doi.org/10.7554/eLife.34176.016

Interestingly, the differences in dissociation rate linked to differences in actin concentrations seemed to disappear when force is applied: the weaker values of $k_{off}$ increased more steeply with force, resulting in a convergence of the dissociation rates when tension was applied (clearly visible for mDia1, in the log-linear representation of *Figure 4C* and *Figure 4—figure supplement 2A*). Likewise, the dissociation constant increased with tension in a seemingly identical fashion whether the filaments were elongating from actin alone or with an excess of profilin (*Figure 4D,F* and *Figure 4—figure supplement 2B*).

We found a similar increase of $k_{off}$ with tension for mDia1 FH2 dimers (i.e. without FH1 domains), and for mDia1 FH1-FH2 dimers anchored via their FH1 or their FH2 domains (i.e. whether force is applied to FH2 alone or to FH1 as well) (*Figure 4E*). These observations indicate that FH1 domains do not participate in the mechanical modulation of formin processivity.

## Discussion

### Processivity mostly relies on FH2-filament interactions, with an unexpected contribution of FH1 domains

Formin control of actin filament elongation at the barbed end is mediated by its homology domains FH1 and FH2, as well as its tail domain, DAD (*Gould et al., 2011*; *Vizcarra et al., 2014*). We quantified formin processivity by measuring its dissociation rate $k_{off}$. Our data indicate that FH2-barbed end interactions are destabilized by ions (*Figure 2A*). These results confirm that salt bridges mediating FH2-actin interactions, which have been predicted from molecular dynamics simulations (*Baker et al., 2015*), are essential determinants of the residence time of formin at the barbed end. Our data also indicate that FH2-actin interactions are destabilized by the presence of a fluorescent label on actin subunits (*Figure 2C*), consistent with the notion that the lateral contacts of FH2 with actin subunits are essential to maintain the formin at the barbed end (*Otomo et al., 2005*).

Unexpectedly, we show here that FH1 domains also contribute to keeping formin at the barbed end (*Figure 3*). Since both FH1 domains and profilin-actin interactions (*Figure 3—figure supplement 2*) are required for profilin to reduce formin dissociation, our results appear in good agreement with the proposition that FH1 delivers profilin-actin to the barbed end by forming a 'ring complex' (*Vavylonis et al., 2006*), where profilin simultaneously interacts with the barbed end and one polyproline track of one of the two FH1 domains. The ring complex is also likely formed when profilin is brought to the barbed end by FH1 without an actin monomer. It seems natural that, in such a configuration, the FH1 domains would constitute an obstacle to the dissociation of the FH2 dimer from the filament barbed end (*Figure 3F*). It is difficult to estimate the frequency at which the ring complex is formed, based on the low affinity of profilin for polyproline tracks (*Perelroizen et al., 1994*) and actin barbed ends (*Pernier et al., 2016*). However, we can assume that the ring complex is formed at least when delivering profilin-actin to the barbed end, and this may be enough to significantly decrease the rate of detachment associated with the addition of new actin subunits (i.e., detachment from the transition state, later discussed and illustrated in *Figure 6*).

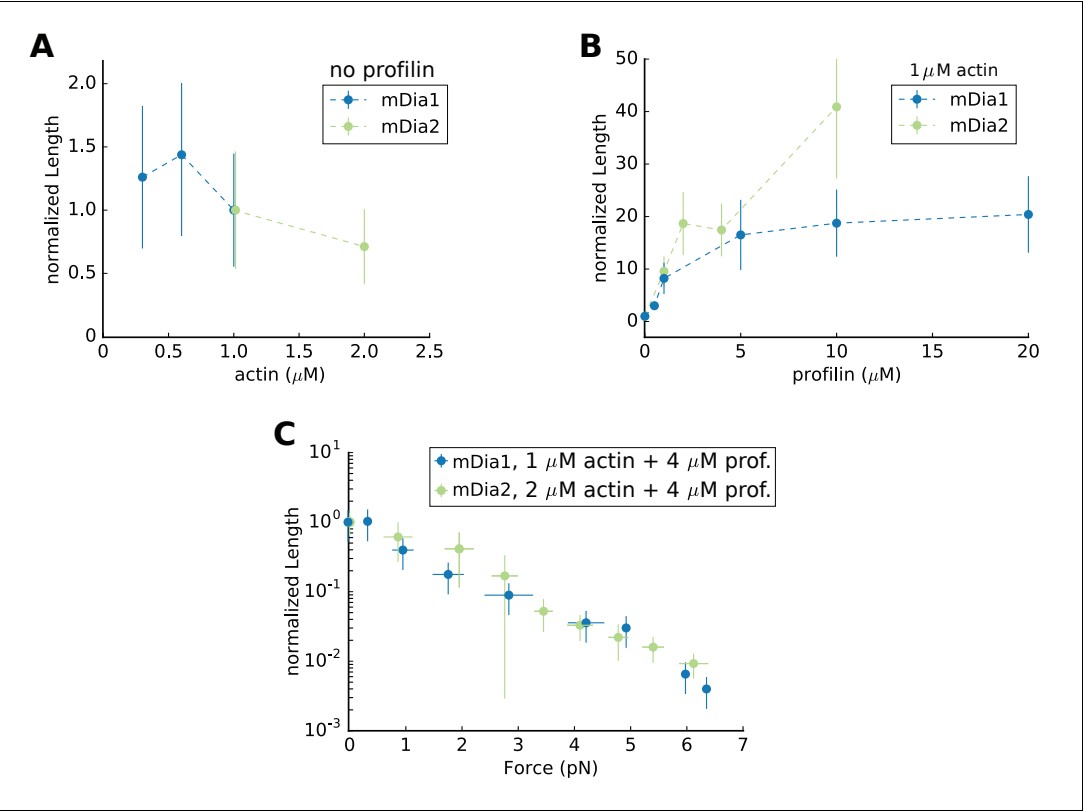

**Figure 5.** Mean Filament Length generated by mDia1 or mDia2 formins. Mean flament length (**A**) as a function of actin concentration, in the absence of profilin, normalized by its value at 1 μM actin (which equals 0.65 μm for mDia1 and 8.22 μm for mDia2); (**B**) mean length as a function of profilin concentration, at 1 μM actin, normalized by its value in the absence of profilin; (**C**) mean length as a function of force, at 2 μM actin 4 μM profilin for mDia2, and at 1 μM actin 4 μM profilin for mDia1, normalized by its value in the absence of force (83.7 μm for mDia2 and 7.45 μm for mDia1). All data were acquired at 100 mM KCl. Error bars indicate standard deviations.

DOI: https://doi.org/10.7554/eLife.34176.018

The following source data is available for figure 5:

**Source data 1.** Spreadsheet containing the data plotted in *Figure 5*.

DOI: https://doi.org/10.7554/eLife.34176.019

---

This contribution of FH1 domains also confers a new function to profilin: not only does it allow a rapid barbed end elongation, it also helps maintain formin at the barbed end. If rapid elongation were to be achieved without profilin, formins would dissociate very rapidly (*Figure 3A,C*).

We have also shown that, when mDia1 FH1 domains were severely truncated, reducing their number of polyproline tracks from 14 to 2, they were still able to perform their tasks regarding both the acceleration of elongation and the reduction of dissociation in the presence of profilin (*Figure 3B*). These observations are consistent with earlier results on yeast formin Bni1p showing that polyproline tracks located closest to the FH2 domain can mediate a robust polymerization rate in absence of the other polyproline tracks (*Courtemanche and Pollard, 2012*; *Paul and Pollard, 2008*).

This FH1-profilin-mediated stabilization does not seem to resist pulling forces, since the formin dissociation rate increases equally fast in the presence of profilin as without profilin (*Figure 4C,D and F*), or even when FH1 domains are absent (*Figure 4E*).

The different factors that we report here to affect formin processivity may also modulate the resulting average filament length (i.e. the filament elongation rate divided by the formin dissociation rate) in a non-trivial way (*Figure 5*). An increase in actin concentration increases the elongation rate while reducing processivity, and as a result the filament length varies moderately (*Figure 5A*). In contrast, increasing the profilin concentration efficiently generates longer filaments (*Figure 5B*) since both processivity and elongation rates are strongly increased. The application of mechanical tension

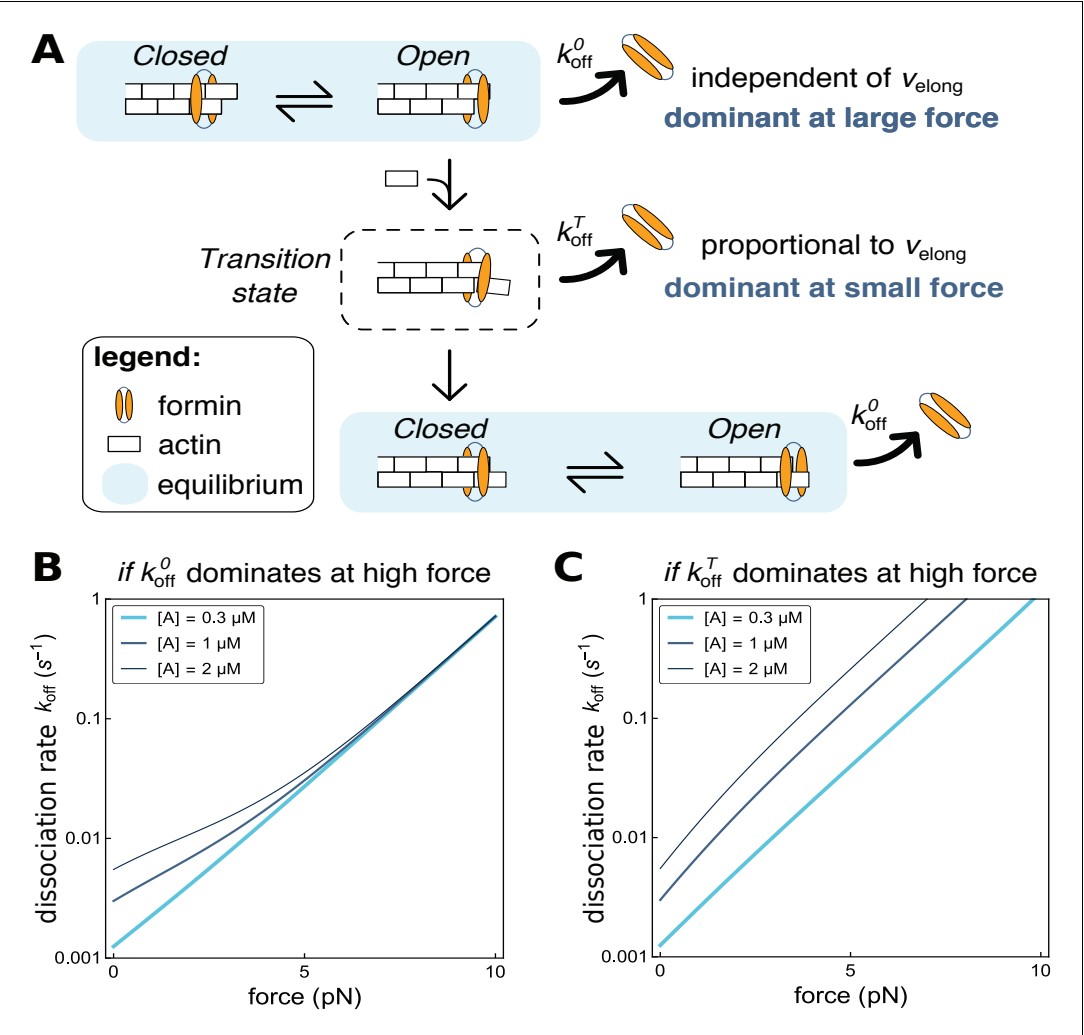

**Figure 6.** Modeling formin dissociation, in the absence of profilin. (A) Sketch summarizing the conformations adopted by the FH2 dimer and the actin filament barbed end in our model, in the absence of profilin (for a complete description of the model, see Appendix 1). The system is in rapid equilibrium between an open and a closed state (depicted here as in the 'stair-stepping' model) and only the open state allows the addition of a new actin subunit at the barbed end. Following this elongation event, the system is in a transition state, which decays rapidly into a new open-closed rapid equilibrium. Formin dissociation from the barbed end can occur while the system is in the open state (with rate $k_{off}^O$) or in the transition state (rate $k_{off}^T$). The global, observable dissociation rate $k_{off}$ comprises these two routes. (B,C) Predictions of the model for the variation of the dissociation rate $k_{off}$ as a function of force, in log-linear representations. In both cases, $k_{off}^T$ is the dominant contribution at zero force. In B, $k_{off}^O$ increases more strongly than $k_{off}^T$ when force is applied and thus becomes dominant at high force (computed with working distances $\delta_O = \delta$ and $\delta_T = 0$, see Appendix 1). In C, $k_{off}^T$ increases more strongly than $k_{off}^O$ when force is applied ($\delta_O = 0$ and $\delta_T = \delta$, see Appendix 1).
DOI: https://doi.org/10.7554/eLife.34176.020

drastically decreases the resulting filament length (*Figure 5C*) because the increase in elongation rate (see *Jégou et al., 2013*) for mDia1, *Figure 4—figure supplement 3* for mDia2) is very moderate compared to the decrease in processivity. For both mDia1 and mDia2, filament length is typically reduced ten-fold every 3 pN. Compared to mDia1, mDia2 elongates filaments more slowly in all the conditions we tested, but it is far more processive, resulting in the formation of much longer filaments. This feature appears to reflect differences in their FH2-DAD domains since it persists in the absence of profilin.

## Modeling FH2 conformations at the barbed end

Putting the contributions of FH1 domains and profilin aside, our results show that the elongation velocity, hence the addition of actin subunits, enhances formin dissociation from the barbed end (*Figure 3A*). This suggests that the FH2 dimer goes through a transient, weakly bound state, every time a new actin subunit is added (*Figure 6A*), as already proposed by (*Paul and Pollard, 2008*). Based on this idea, we have built a mathematical model predicting the elongation velocity $v_{elong}$ and dissociation rate $k_{off}$ for a barbed-end associated formin as a function protein concentrations and force. This model and its predictions are presented in detail in Appendix 1.

As our model includes a substantial number of chemical reactions and associated reaction rates, we focus less on obtaining precise fits to the experimental data - which are somewhat trivial and uninformative when a large number of adjustable parameters are involved - and instead demonstrate that the *qualitative* shape of the curves predicted by our model is consistent with our experimental measurements. This shows that the agreement between our model and the data is essential, and not an accident of a specific set of values for the fitting parameters.

Our model, while it does not attempt to explicitly describe the details of FH1 activity, as done by Vavylonis and colleagues (*Vavylonis et al., 2006*), does include an effective affinity of profilin for barbed ends, and is able to account for our experimental data on profilin by simply considering that the presence of profilin at the barbed end blocks formin dissociation (Appendix 1 and figures therein). Our model thus ties together our observations in a global, consistent description. It also provides insights into the FH2 dimer conformations and the effect of applied tension, which we now summarize here.

Structural details of Bni1p(FH2)-actin interactions (*Otomo et al., 2005*) have led to the proposal that, as they wait for the addition of a new actin subunit, the FH2 dimer and the barbed end are in a rapid equilibrium between an elongation-competent 'open' state and an elongation-forbidding 'closed' state. In the frame of the subsequently proposed 'stair-stepping' model, FH2 hemidimer translocation (along the filament's main axis and over a distance of one actin monomer size) is associated to this rapid equilibrium. In contrast, the 'stepping-second' model proposes that the open-closed equilibrium involves no such FH2 hemidimer translocation, which would instead take place after each subunit addition and thus be related to the aforementioned transition state (*Paul and Pollard, 2008*).

Our earlier work showing that tension accelerates mDia1-mediated elongation (*Jégou et al., 2013*) and more recent work applying tension with magnetic tweezers (*Yu et al., 2017*), both indicate that the open-closed equilibrium corresponds to a working distance of one monomer size, consistent with the stair-stepping model. We have thus chosen this model for our schematic representations of the open-closed equilibrium (*Figure 6*), even though our data on formin dissociation does not favor one model over the other. The conclusions we draw from our present model of formin dissociation do not require the stair-stepping context.

Our model for dissociation primarily includes the notion that the FH2 dimer goes through a transient, dissociation-prone conformation every time a new actin subunit is added. As sketched in *Figure 6A*, formin can thus dissociate following two routes: (1) the FH2 dimer unbinds from the barbed end from the open state during its rapid open-closed equilibrium, with a rate $k_{off}^{O}(f)$, or (2) the FH2 dimer unbinds during the transition state that follows subunit addition, with a rate $k_{off}^{T}(f)$.

In the absence of force, our data show a strong dependence of formin dissociation on actin concentration, i.e. on elongation rate (*Figure 3A and C*) meaning that $k_{off}^{T}(f = 0)$ is the dominant contribution to the global $k_{off}(f = 0)$. When pulling forces are applied, the formin dissociation rates for different actin concentrations converge, *i.e.* $k_{off}(f)$ does not depend on actin concentration anymore (*Figure 4C and F*). The model predicts such a behavior when $k_{off}^{O}(f)$ increases with force more strongly than $k_{off}^{T}(f)$, and thus becomes dominant at high forces (*Figure 6B*). In contrast, the situation where $k_{off}^{T}(f)$ remains the dominant contribution to dissociation results in curves for $k_{off}(f)$ at different actin concentrations that remain well separated at high forces (*Figure 6C*). Together our model and data thus indicate that, while dissociation from the transition state is the dominant route at low force, it is the dissociation from the open state that dominates at high force.

## How do cells manage formin dissociation in a mechanical context?

Our results show that mechanical tension plays a dominant role in the modulation of formin processivity. The dramatic enhancement of formin dissociation, upon application of piconewton forces, appears difficult to compensate with the other factors we have tested, such as actin and profilin concentrations. In cells, where filaments are likely to be tensed mainly because of myosin activity, our results raise questions regarding how these filaments may remain in interaction with membrane-anchored formins. Since it seems unlikely that filaments detach from membranes as soon as moderate forces are applied, they may cumulate alternative anchoring strategies, or see their interaction with formins reinforced by other factors.

In cells, formin-elongated filaments are often found in bundles, a situation which could allow dissociated formins to rapidly rebind to barbed ends. Also, recent studies have shown that regulatory proteins could directly bind to formins and modulate their activity (*e.g.*, Ena/VASP (*Bilancia et al., 2014*), CLIP170 (*Henty-Ridilla et al., 2016*), or Spire/FMN2 interactions (*Montaville et al., 2014*)). The stabilization of formin-filament interactions in a mechanical context by such proteins is a hypothesis that should be addressed in future experiments.

# Materials and methods

### Key resources table

| Reagent type (species) or resource | Designation | Source or reference | Identifiers | Additional information |
|---|---|---|---|---|
| Strain, strain background (*E. coli*) | BL21(DE3) | Thermo Fischer | Cat# C600003 | |
| Biological sample (Rabbit) | Rabbit muscle | INRA Jouy-en-Josas | N/A | |
| Peptide, recombinant protein | Mouse mDia1(FH1-FH2-DAD) | Uniprot | O08808 | seq. 552–1255 aa |
| Peptide, recombinant protein | Mouse mDia2(FH1-FH2-DAD) | Uniprot | Q9Z207 | seq. 521–1171 aa |
| Peptide, recombinant protein | Human profilin-1 | Uniprot | P07737 | |
| Antibody | PentaHis Biotin conjugate | Qiagen | 34440 | |
| Chemical compound, drug | Alexa Fluor 488 succinimidyl ester | Life Technologies | Cat#A20000 | |
| Commercial assay or kit | Protino Ni-NTA Agarose beads | Macherey-Nagel | Cat#745400.25 | |
| Commercial assay or kit | HiLoad 16/60 Superdex 200 gel filtration column | GE Healthcare | Cat#28-9893-35 | |
| Software, algorithm | numpy/scipy packages | Python | | |

### Proteins and buffers

Skeletal muscle actin was purified from rabbit muscle acetone powder (Pel-freeze) following the protocol described in (*Wioland et al., 2017*), adapted from the original protocol (*Spudich and Watt, 1971*). Actin was fluorescently labeled on accessible surface lysine 328 of F-actin (*Tóth et al., 2016*), using Alexa 488-NHS (LifeTechnologies).

Recombinant mouse formins mDia1(SNAP-FH1-FH2-DAD-6xHis) and mDia2(SNAP-FH1-FH2-DAD-6xHis) were expressed in E. Coli Rosetta 2 (DE3) and purified following the protocol described in (*Romero et al., 2004*).

Recombinant human profilin I was expressed in E. Coli BL21 Star (DE3) and purified following the protocol described in details in *Wioland et al. (2017)*, based on the original protocol by *Gieselmann et al. (2008)*.

Spectrin-actin seeds were purified from human erythrocytes as described in *Wioland et al. (2017)*, based on the original protocol by *Casella et al. (1986)*.

Experiments were performed in F-buffer (5 mM Tris-HCl pH 7.8, 1 mM MgCl2, 0.2 mM EGTA, 0.2 mM ATP, 10 mM DTT and 1 mM DABCO) with various concentrations of KCl, as indicated in the main text and figures.

## Microfluidics setup and experiments

Protein solutions were injected into a Poly-Dimethyl-Siloxane (PDMS, Sylgard) chamber, 20 μm or 40 μm in height, 800 μm in width and 1 cm in length. Chambers were mounted on glass coverslips previously cleaned for 20 min in ultrasonic baths of 1M KOH, ethanol and dH20. PDMS chambers and glass coverslips were UV-treated (UVO cleaner, Jelight) to allow them to bind tightly to each other. We used cross-shaped channels with three inlets. We controlled the pressure in the reservoir and measured the flow rate in each channel using an MFCS and Flow Units (Fluigent).

For experiments with anchored pointed ends (configurations shown in *Figure 1A,B*) the chamber was first filled with F-buffer without KCl. We then injected actin-spectrin seeds, 10 pM for 5 min, which adsorbed to the glass surface non-specifically. The surface was then passivated with 5% bovine serum albumin for at least 10 min.

The anchoring of formins to the coverslip surface (configurations shown in *Figures 1C* and *4A*) was achieved in various ways, with similar results. Surfaces were first passivated and functionalized with biotin, either with PLL-PEG containing a fraction of PLL-PEG-biotin (SuSoS, Switzerland) or with a mixture of BSA and biotinylated BSA. The surfaces were then incubated for 5 min with neutravidin (20 μg/mL) and rinsed. The various formin constructs all contained a C-terminal 6xHis tag to anchor them via a biotinylated anti-His (penta-His, Qiagen). To anchor specifically the mDia1 (FH1-FH2-DAD) via its N-terminus, we used a biotinylated SNAP-tag construct.

## Microscopy and image acquisition

The microfluidic setup was placed on a Nikon TiE inverted microscope, equipped with a 60x oil-immersion objective. We either used TIRF, HiLo or epifluorescence depending on the background fluorophore concentration in solution. Two different TiE microscope setups were used. The TIRF setup was controlled by Metamorph, illuminated in TIRF or epifluorescence by 100 mW tunable lasers (iLAS2, Roper Scientific), and images were acquired by an Evolve EMCCD camera (Photometrics). The other TiE setup was controlled by micromanager (*Edelstein et al., 2014*), illuminated with a 200W Xcite lamp (Lumen dynamics) and images were acquired by an sCMOS Orca-Flash4.0 V2 + camera (Hamamatsu).

Images were analyzed using ImageJ software.

The experiments were performed at room temperature, in an air-conditioned environment. We nonetheless measured day-to-day variations of room temperature, between 19°C and 23°C, and found that these temperature changes correlated with variations in filament elongation rates and formin dissociation rates: higher temperatures favored faster elongation and faster dissociation. To minimize the impact of such variations, and obtain consistent data, experiments and their controls were systematically repeated on the same day.

## Data analysis

To avoid any bias related to the selection of filaments during analysis, a rectangular region containing a few tens of filaments was randomly chosen in the microscope field of view, and all the filaments in this region were analyzed. Within this population, filaments were excluded from our analysis only in the following specific cases. We excluded filaments whose ends were difficult to locate because they overlapped with other filaments. We also excluded filaments that sometimes seemed to stick to the surface or, in the case of experiments with anchored formins, appeared to stall (see Supp. Movies).

Movies were analyzed with ImageJ. The Subtract Background plugin was sometimes used to enhance the contrast, with a rolling ball radius of 50 pixels.

## Quantifying formin dissociation rates and their error bars, in the absence of force

For each experiment, from a randomly chosen filament population, we noted the time at which each individual formin dissociated from the barbed end. The survival fraction of each experiment was then fitted by a single exponential using the numpy/scipy numerical packages of Python.

In order to quantify the statistical uncertainty in the estimation of the dissociation rate $k_{off}$ resulting from the exponential fits of the survival fractions $S(t)$ (shown for example in *Figures 1,2A*; *Figure 2C*), we performed numerical simulations of the experiment (using Python). The program simulated a large number ($M$ = 10,000) of experiments, each consisting in $N$ filaments randomly losing their formin with rate constant $k_0$. The survival fraction of each simulated experiment was fitted by a single exponential, resulting in the generation of $M$ estimated rates $k_{est}$. The distribution of these $k_{est}$, centered on $k_0$, allowed us to compute the width of the confidence intervals. We could thus verify that a 65% confidence interval corresponded to errors of approximately $k_0/N^{0.5}$.

## Analysis of experiments with striped filaments

Our standard experiment (*Figure 1A*) relied on the ability to image filaments and on the acceleration of their elongation by formins in order to assess their presence at the barbed end. In order to determine the elongation velocity and the formin dissociation rate in conditions where actin could not be directly imaged (i.e. unlabeled actin) and/or when the presence of formin was not readily detected by a change in elongation velocity (i.e. in the absence of profilin), other configurations were used. A possible alternative was to anchor the formins to the coverslip surface and work with low forces (*Figure 1C*). In order to obtain results with unanchored formins and zero force, we have used a 'striped filaments' protocol (illustrated in *Figure 1B*). It consisted in exposing filaments to alternating conditions: a duration $\Delta t_1$ with condition 1 (the condition of interest, with unknown elongation rate $v_1$ and formin dissociation rate $k_1$), and a duration $\Delta t_2$ with condition 2 (containing profilin and labeled actin, with predetermined elongation rate $v_2$ and formin dissociation rate $k_2$). The resulting, striped filament population was imaged at interval ($\Delta t_1 + \Delta t_2$) and had a measurable elongation rate $v = (\Delta t_1 v_1 + \Delta t_2 v_2)/(\Delta t_1 + \Delta t_2)$ and formin dissociation rate $k = (\Delta t_1 k_1 + \Delta t_2 k_2)/(\Delta t_1 + \Delta t_2)$. Knowing $v_2$ and $k_2$, we could thus determine $v_1$ and $k_1$. The results we obtained were consistent with those from experiments with anchored formins, at very low force.

## Analysis of experiments with pulling forces

We measured the fraction $S(t)$ of filaments growing from surface-anchored formins that remained attached over time, while force was applied on the filaments by viscous drag. The observed filament detachment rate $k_{obs}(t) = (dS/dt)/S(t)$ increases over time, as the filaments get longer and the average force exerted on them thus increases. This force has been calibrated (*Jégou et al., 2013*) and we can compute the average force $f(t)$ exerted on the population of filaments, homogeneous in length.

An important point is to verify whether the filament detachment events that we observe during our experiment do correspond to formin-filament dissociation events. We thus sought to estimate what percentage of the monitored formins were still present and functional at the end of an experiment. To do so, following the experiment, we exposed the surface to a solution of actin to test which formins could nucleate new filaments. We observed that ~74% of formins were still present and able to nucleate filaments during this test (*Figure 4—figure supplement 1*), regardless of the force applied during the experiment (between 0 and 6 pN). This indicated that at least 74% of the formins monitored during the experiment were still anchored and functional when their filament was observed to detach from the surface. The measured filament detachment rate $k_{obs}$ thus reflected the formin dissociation rate $k_{off}$ within a reasonable error: $0.74 \, k_{obs} < k_{off} < k_{obs}$ (corresponding to the vertical error bars shown in *Figure 4—figure supplement 2*).

We could thus plot the formin dissociation rate $k_{off}$ as a function of the applied force $f$. Each individual experiment generated a survival fraction $S(t)$ (as in *Figure 4B*) from which we deduced a number of points $k_{off}(f)$, as shown in *Figure 4—figure supplement 2*. The horizontal error bars indicate the standard deviations in f, based on the length dispersion of the filaments. Experiments carried out with different microfluidics flow rates explored different ranges of force, with some overlap between experiments. For clarity, data points were grouped in bins of similar force, and averaged.

The resulting plots are shown in *Figure 4*, where the error bars indicate the standard deviations for $f$ and for $k_{off}$ within each bin.

## Acknowledgements

The authors thank David Kovar and Dennis Zimmermann for the plasmids of formin mDia2 and mutant profilin-R88E. The authors acknowledge funding from the Fondation pour la Recherche Médicale (grant to GRL and ML), the Human Frontier Science Program (grant RGY0066 to GRL), the H2020 European Research Council (grant StG-679116 to AJ, grant StG-677532 to ML) and the Fondation ARC pour la Recherche sur le Cancer (postdoctoral fellowship to HW).

## Additional information

### Funding

| Funder | Grant reference number | Author |
|---|---|---|
| Fondation ARC pour la Recherche sur le Cancer | Postdoctoral fellowship | Hugo Wioland |
| Fondation pour la Recherche Médicale | DEI20151234415 | Martin Lenz Guillaume Romet-Lemonne |
| H2020 European Research Council | StG-677532 | Martin Lenz |
| Human Frontier Science Program | RGY0066 | Guillaume Romet-Lemonne |
| H2020 European Research Council | StG-679116 | Antoine Jegou |

The funders had no role in study design, data collection and interpretation, or the decision to submit the work for publication.

### Author contributions

Luyan Cao, Resources, Formal analysis, Validation, Investigation, Visualization; Mikael Kerleau, Emiko L. Suzuki, Formal analysis, Validation, Investigation, Visualization; Hugo Wioland, Software, Formal analysis, Funding acquisition; Sandy Jouet, Berengere Guichard, Resources, protein purification and characterization; Martin Lenz, Software, Formal analysis, Funding acquisition, Validation, Methodology, Writing—original draft; Guillaume Romet-Lemonne, Antoine Jegou, Conceptualization, Formal analysis, Supervision, Funding acquisition, Validation, Investigation, Visualization, Methodology, Writing—original draft, Project administration, Writing—review and editing

### Author ORCIDs

Hugo Wioland http://orcid.org/0000-0001-5254-9642
Martin Lenz http://orcid.org/0000-0002-2307-1106
Guillaume Romet-Lemonne http://orcid.org/0000-0002-4938-1065
Antoine Jegou http://orcid.org/0000-0003-0356-3127

### Decision letter and Author response

Decision letter https://doi.org/10.7554/eLife.34176.026
Author response https://doi.org/10.7554/eLife.34176.027

## Additional files

### Supplementary files

• Transparent reporting form
DOI: https://doi.org/10.7554/eLife.34176.021

## Data availability

Source data files are microscope movies. Three are provided as Videos 1-3 and more can be shared upon request to the corresponding authors (romet@ijm.fr or antoine.jegou@ijm.fr). The totality of the files has not been made available at this point due to the size and volume of the raw data and the extensive level of detail (frame rate, pixel size, change of chemical conditions, etc.) that would be required in order for each movie to be exploitable.

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

## Appendix 1

DOI: https://doi.org/10.7554/eLife.34176.022

# Mathematical modeling

Here we describe a mathematical model predicting the elongation velocity $v_{elong}$ and dissociation rate of a barbed-end associated formin as a function of actin and profilin concentrations $c_a$ and $c_p$, as well as the force $f$ applied to the formin in the direction of filament elongation. We present the model in Model description and derive its general predictions in Elongation and dissociation rates. We then specialized these results to the measurements performed in the main text in Specific predictions.

As our model includes a substantial number of chemical reactions and associated reaction rates, we focus less on obtaining precise fits to the experimental data—which are somewhat trivial and uninformative when a large number of adjustable parameters are involved—and instead demonstrate that the *qualitative* shape of the curves predicted by our model is always consistent with our experimental measurements. This demonstrates that the agreement between our model and the data is essential, and not and accident of a specific set of values for the fitting parameters.

# Model description

The model is a kinetic description of the barbed end-formin complex based on transitions between three basic states. Following (*Otomo et al., 2005*), we assume that formin can be associated with the filament barbed end in either a "closed" or an "open" conformation, of which only the latter allows for further filament elongation. These two states, henceforth abbreviated as $C$ and $O$ are assumed to be in rapid equilibrium, implying that their probabilities $P_c$ and $P_o$ are constrained by the detailed balance condition

$$\frac{P_O}{P_C} = \frac{\exp(-\beta\varepsilon_O + \beta f\delta)}{\exp(-\beta\epsilon_C)} = \exp(-\beta\epsilon + \beta f\delta), \tag{A1}$$

where $\beta = \frac{1}{k_B}T$ is the inverse thermal energy, $\epsilon = \epsilon_O - \epsilon_C$ is the energy difference between states $O$ and $C$ and $\delta$ is the average distance over which the formin moves along the filament as it transitions from $C$ to $O$. Only state $O$ allows the recruitment of a new actin monomer to the barbed end, which happens irreversibly with a rate $k_a c_a$ proportional to the actin concentration in solution. This monomer addition takes the system to a short-lived transient state, denoted by $T$. As schematized in *Appendix 1—figure 1(a)*, this state decays with a rate $\frac{1}{\tau}$ into a new fast $C \rightleftharpoons O$ equilibrium with one more actin monomer, implying that the new $C$ state is shifted with respect to the original one by a $\delta$ distance, and similarly for $O$.

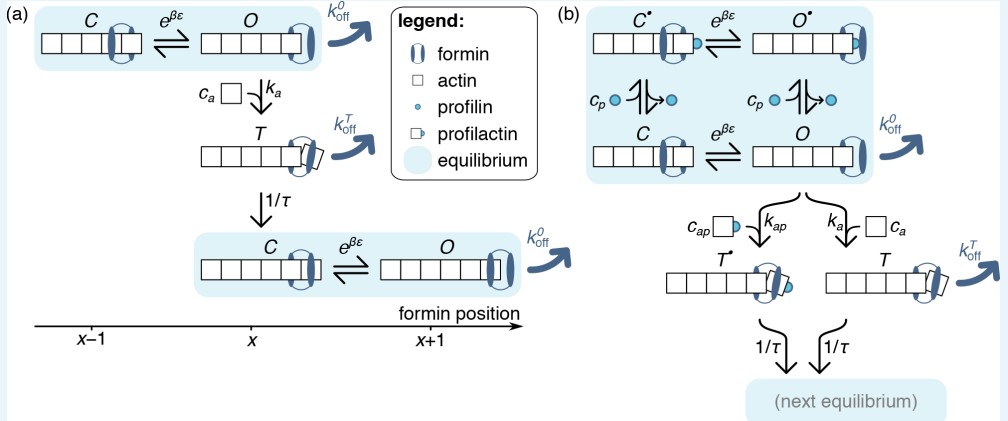

**Appendix 1—figure 1.** Model of formin function based on transitions between discrete states,

as described in the text. Unlike those of the main text, the schematics presented here only picture one actin protofilament for simplicity, without any implications for the model itself. In both panels, formin dissociation from states $O$ and $T$ is indicated by thick dark blue arrows. The FH1 domains, which lead to the formation of a ring complex that hampers this dissociation in profilin-associated states, is not explicitly represented here. (**a**) Simplified, no-profilin model introducing the notion of rapid equilibrium between monomer additions, and transit through a short-lived state $T$ upon monomer addition. (**b**) Model taking into account the association of profilin with the filament barbed end.

DOI: https://doi.org/10.7554/eLife.34176.023

While the formin is bound to the actin in all three states, thermal agitation and the force $f$ may pull it to the right and off the filament, implying formin dissociation. This may happen in state $O$ or $T$, but not in state $C$, reflecting the fact that a formin starting from state $C$ must first go through $O$ before it can leave the filament. Denoting by $\delta_O$ and $\delta_T$ the distance over which the formin must be pulled to be ripped off the filament when in state $O$ or $T$ respectively, we assimilate the dissociation process to a simple Kramers escape problem and write the associated dissociation rates

$$k_{off}^O(f) = k_{off}^O(O) \times e^{\beta j \delta_0}, \;\; k_{off}^T(f) = k_{off}^T(O) \times e^{\beta j \delta_T}. \tag{A2}$$

To account for the possibility of profilin-actin association, we introduce profilin-associated versions of each of the aforementioned states, which we denote as $C^\bullet$, $O^\bullet$ and $T^\bullet$. In these states, the last actin on the filament barbed end is bound to a profilin, which sterically prevents the addition of any new actin monomer to the filament prior to its detachment. We denote by $K_d$ the equilibrium dissociation constant of the following reaction

$$(\text{profilinated barbed end}) \rightleftharpoons (\text{barbed end}) + (\text{profilin}). \tag{A3}$$

Combined with the assumption that states $C^\bullet$ and $O^\bullet$ are at a rapid equilibrium with states $C$ and $O$, this implies

$$\frac{P_{C^\bullet}}{P_C} = \frac{P_{O^\bullet}}{P_O} = \frac{c_p}{K_d}, \tag{A4}$$

which we combine with **Equation (A1)** and the normalization condition $P_{C^\bullet} + P_C + P_{O^\bullet} + P_O = 1$ to obtain

$$P_{C^\bullet} = \frac{e^{\beta \epsilon} c_p / K_d}{\left(1 + c_p / K_d\right)\left(e^{\beta \epsilon} + e^{\beta f \delta}\right)} \tag{A5a}$$

$$P_C = \frac{e^{\beta \epsilon}}{\left(1 + c_p / K_d\right)\left(e^{\beta \epsilon} + e^{\beta f \delta}\right)} \tag{A5b}$$

$$P_{O^\bullet} = \frac{e^{\beta f \delta} c_p / K_d}{\left(1 + c_p / K_d\right)\left(e^{\beta \epsilon} + e^{\beta f \delta}\right)} \tag{A5c}$$

$$P_O = \frac{e^{\beta f \delta}}{\left(1 + c_p / K_d\right)\left(e^{\beta \epsilon} + e^{\beta f \delta}\right)} \tag{A5d}$$

State $T^\bullet$, on the other hand, can only be reached by adding a profilactin to a filament barbed end in the $O$ state, which occurs with rate $k_{ap} c_{ap}$, where $c_{ap}$ denotes the profilactin concentration in solution. Similar to the behavior of state $T$, we assume that state $T^\bullet$ quickly transitions into a new $C^\bullet \rightleftharpoons C \rightleftharpoons O \rightleftharpoons O^\bullet$ equilibrium with a rate $1/\tau$.

As discussed in the main text, we assume that none of the profilinated states is amenable to formin dissociation, as the interactions between formin's FH1 domain and the filament-bound profilin helps stabilize its attachment to the filament.

## Elongation and dissociation rates

To compute the filament's average elongation rate, we compute the average time $1/v_{\text{elong}}$ required to add a monomer to it. When in the $C^\bullet \rightleftharpoons C \rightleftharpoons O \rightleftharpoons O^\bullet$ equilibrium, the system spends a fraction $P_O$ of its time in the $O$ state. During this time, it may transition into the $T$ and $T^\bullet$ states with respective rates $k_a c_a$ and $k_{ap} c_{ap}$, implying an overall escape rate out of the equilibrium of $k_a c_a P_O + k_{ap} c_{ap} P_O$. Following our assumption that states $T^\bullet$ and $T$ are short-lived, the time scale $\tau$ is negligible in front of the escape time and thus the elongation velocity (measured in number of monomers per unit time) reads

$$v_{elong} = k_a c_a P_O + k_{ap} c_{ap} P_O. \tag{A6}$$

Our model allows for two sources of formin dissociation. First, formin may leave the filament while in the $O$ state. As formin spends a fraction $P_O$ of its time in this state, the associated dissociation rate reads $k_{\text{off}^O(f)P_O}$. Second, formin may leave the filament while in the $T$ state. While this state is very transient, it has been argued that it is also highly unstable (**Paul and Pollard, 2008**) and thus that the associated dissociation rate may be significant. To estimate this dissociation rate, we first consider a system that has just transitioned into the $T$ state. The system may escape this state through either one of two mechanisms, namely a transition into the $C^\bullet \rightleftharpoons C \rightleftharpoons O \rightleftharpoons O^\bullet$ equilibrium (with rate $1/\tau$), or dissociation [with rate $k_{\text{off}^T(f)}$]. Since both of these rates are constant over time, it is easy to show that the probability to escape the $T$ state through dissociation is $k_{\text{off}^T(f)\tau}/\left[1 + k_{\text{off}^T(f)\tau}\right]$. Similarly, starting from the $C^\bullet \rightleftharpoons C \rightleftharpoons O \rightleftharpoons O^\bullet$ equilibrium the probability of entering the $T$ state rather than the $T^\bullet$ state reads $k_a c_a / (k_a c_a + k_{ap} c_{ap})$, implying that the probability of losing the formin while transitioning from one equilibrium to the next reads

$$\frac{k_a c_a}{k_a c_a + k_{ap} c_{ap}} \times \frac{k_{\text{off}^T(f)\tau}}{1 + k_{\text{off}^T(f)\tau}} \tag{A7}$$

Finally, as there is on average one such transition per time interval of duration $1/v_{\text{elong}}$, the overall dissociation rate of the formin reads

$$\begin{aligned} k_{off} &= k_{off}^O(f) P_O + v_{elong} \frac{k_a c_a}{k_a c_a + k_{ap} c_{ap}} \frac{k_{off}^T(f)\tau}{1 + k_{off}^T(f)\tau} \\ &= \frac{e^{\beta f \delta}}{e^{\beta \epsilon} + e^{\beta f \delta}} \frac{1}{1 + c_p/K_d} \left[ k_{off}^O(f) + k_a c_a \frac{k_{off}^T(f)\tau}{1 + k_{off}^T(f)\tau} \right], \end{aligned} \tag{A8}$$

where the first and second terms in the square brackets relate to the dissociation rate in the open and transient state, respectively.

## Specific predictions

Here we specialize the results of **Equations (A6) and (A8)** to experimentally relevant situations, showing robust agreement with the data of the main text. In the following we make the simplifying assumption that the equilibrium dissociation constant $\simeq 0.1\,\mu\text{M}$ of the chemical equilibrium

$$\text{G} - \text{profilactin} \rightleftharpoons \text{G} - \text{actin} + \text{profilin} \tag{A9}$$

in solution is much smaller than the other relevant concentrations in the system (typically a few $\mu\text{M}$), or equivalently that an excess of profilin in solution with respect to actin implies that essentially all actin is associated with profilin, with a negligible concentration of residual non-associated actin. Denoting by [A] and [P] the nominal concentrations of actin and profilin initially introduced in the solution, this implies

$$c_a = \begin{cases} [A]-[P] & \text{if}[A]>[P] \\ 0 & \text{if}[A]<[P] \end{cases}, \quad c_p = \begin{cases} 0 & \text{if}[A]>[P] \\ [P]-[A] & \text{if}[A]<[P] \end{cases}, \text{ and } c_{ap} = \begin{cases} [P] & \text{if}[A]>[P] \\ [A] & \text{if}[A]<[P] \end{cases} \quad \text{(A10)}$$

Using this assumption, in the following sections we derive theoretical predictions corresponding to the three main experimental curves of the main text.

## Profilin concentration dependence of the elongation velocity

Plugging **Equation (A10)** into **Equation (A6)**, we obtain

$$v_{elong} = \begin{cases} \frac{k_a[A]+(k_{ap}-k_a)[P]}{1+e^{\beta\epsilon}} & \text{if}[A]>[P] \\ \frac{k_{ap}[A]}{1+e^{\beta\epsilon}}\frac{1}{1+([P]-[A])/K_d} & \text{if}[A]<[P] \end{cases} \quad \text{(A11)}$$

We represent this function in **Appendix 1—figure 2(a)**. Qualitatively, at low profilin the monomer addition rate is modest, with its pace set by the actin addition rate through the $T$ pathway. As the profilin concentration is increased, the availability of profilactin subunits increases, leading to elongation with the faster rate $k_{ap}$ through the $T^\bullet$ pathway. As the profilin concentration [P] exceeds the actin concentration [A], excess profilin accumulates in the solution, shifting the rapid equilibrium of **Appendix 1—figure 1** towards the $C^\bullet$ and $O^\bullet$ states, thus depleting the addition-competent $O$ state and slowing down elongation.

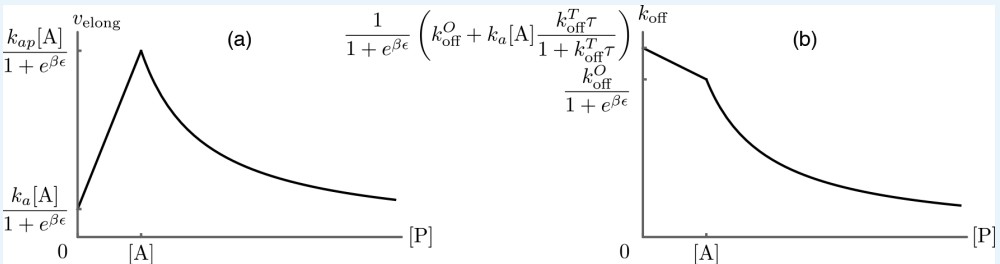

**Appendix 1—figure 2.** Predictions for the elongation velocity $v_{\text{elong}}$ (**a**) and the formin dissociation rate $k_{\text{off}}$ (**b**) as functions of the profilin concentration within the approximation of strong actin-profilin binding of Eq.**(A13)**. While the exact position of the curves is dependent on the choice of parameters as indicated on the figure, their qualitative shapes are a robust prediction of the model, and agree well with the experimental data presented in **Figure 3B** of main text.

DOI: https://doi.org/10.7554/eLife.34176.024

## Profilin concentration dependence of the dissociation rate

Plugging **Equation (A10)** into **Equation (A8)**, we obtain

$$k_{off} = \begin{cases} \frac{1}{1+e^{\beta\epsilon}}\left[k_{off}^O + k_a([A]-[P])\frac{k_{off}^T\tau}{1+k_{off}^T\tau}\right] & \text{if}[A]>[P] \\ \frac{k_{off}^O}{1+e^{\beta\epsilon}}\frac{1}{1+([P]-[A])/K_d} & \text{if}[A]<[P] \end{cases} \quad \text{(A12)}$$

which we plot in **Appendix 1—figure 2(b)**. Qualitatively, the formin dissociation rate is maximal at low profilin concentration, where all monomer additions occur through the dangerous $T$ pathway. As [P] increases, an increasing number of $T$ transitions are replaced by the safe $T^\bullet$ transitions, until at $[P]=[A]$ the $T$ transitions are entirely abrogated. At this and higher concentration, the only remaining cause of formin dissociation is through the $O$ state, and as the profilin concentration is increased above [A], the occupancy of the $O$ state decreases as described in Profilin concentration dependence of the elongation velocity, leading to a further decrease of the dissociation rate.

## Force dependence of the dissociation rate

To describe the force dependence of the formin dissociation rate, we introduce the force-dependent dissociation laws of **Equation (A2)** into the dissociation rate of **Equation (A8)** at

$[P] = 0$. Based on our experimental observations, we restrict our discussion to situations where the formin stays bound to the barbed end for a number of monomer addition steps that is much larger than one, and thus to the regime $k^T_{off}(f)\tau \ll 1$, yielding

$$k_{\text{off}} = \frac{e^{\beta f \delta}}{e^{\beta \epsilon} + e^{\beta f \delta}} \left[ k^O_{off}(0)e^{\beta f \delta_O} + k_a[\text{A}]k^T_{off}(0)e^{\beta f \delta_T}\tau \right] \tag{A13}$$

The two terms in the parenthesis of **Equation (A8)** respectively correspond to dissociation from the $O$ and from the $T$ state. While both rates can contribute at small forces, for large forces the dominant contributor to the dissociation rate will be the process with the largest length scale $\delta_X$ (with $X = O$ or $T$), i.e., dissociation through $O$ if $\delta_O > \delta_T$, or dissociation through $T$ if $\delta_T > \delta_O$. In the former case, the large-force asymptotic dissociation rate $k_{\text{off}} \sim k^O_{off}(0)e^{\beta f \delta_O}$ will be independent of the actin concentration, while in the latter $k_{\text{off}} \sim k_a[\text{A}]k^T_{\text{off}}(0)e^{\beta f \delta_T}\tau$ is proportional to it. As discussed in the main text, the dissociation *vs.* force curves for different actin concentrations converge at large force, indicating that the former hypothesis is correct, *i. e.*, that the force dependence of the dissociation rate in the $O$ state is significantly larger than that in the $T$ state. The corresponding theoretical curves are shown in **Figure 5** of the main text.

