## [Decision Letter]

Thank you for submitting your article "Modulation of formin processivity by profilin and mechanical tension" for consideration by *eLife*. Your article has been favorably evaluated by Anna Akhmanova (Senior Editor) and three reviewers, one of whom is a member of our Board of Reviewing Editors. The reviewers have opted to remain anonymous.

The reviewers have discussed the reviews with one another and the Reviewing Editor has drafted this decision to help you prepare a revised submission.

Summary:

Kerleau and colleagues use TIRF microscopy to monitor formin-mediated actin filament elongation coupled with microfluidic flow to address the impact of profilin, formin FH1 domains, elongation rates and force on the dissociation rate of formin from actin filament barbed ends. Although faster polymerization tends to increase the rate of formin dissociation from filament ends, introduction of profilin into the reaction allows formin to remain associated with growing ends for a longer period of time. Notably, application of tensile force overcomes this stabilizing effect of profilin. To interpret their experimental results, the authors introduce a computational model with two potential pathways for the dissociation of formin from a filament end. Based on this model, they conclude that one of the pathways is favored in the absence of tension, whereas the other is preferred when force is applied to formin-bound filaments.

The work is expertly performed and analyzed, and the results are clearly described. Formin processivity is a key feature of formin function that is currently under-explored and not well understood. The results described in this paper will thus significantly add to the field's understanding of the mechanism of formin-mediated actin polymerization.

Essential revisions:

There is quite a bit of variability in the specific rate constants between diverse formins. The impact of this investigation would certainly be enhanced if additional formins were assayed. The authors could study in particular the effect of profilin and force on the dissociation rate of another formin.

1) It would be extremely helpful for the clarity of the study to plot the survival fraction (of formin) as a function of actin subunits added, instead of as a function of time. This would immediately take into account differences in actin filament elongation rate that occur under the large array of conditions used in the study. This information is in the manuscript with regards to plots of dissociation rate as a function of elongation rate, but perhaps would make the data easier to understand. Furthermore, I believe plotting as a function of actin subunits added (instead of time) would allow for a better baseline with which to compare mDia1 (this study) and other studies that come out with other formins that have vastly different elongation rates over different conditions. This would also take into account changes in elongation rate caused by force (Figure 4).

2) According to the Figure 1, actin filaments are around 30 µm. This is very long compared with the cellular context. How the length of the filament will affect the main conclusions presented here, especially on the tension effect?

3) It would be helpful if the authors included a description of how they fit their survival plot data. I presume that they fit single-exponential decays to their data, but this is not explicitly stated in the text.

4) Since the gating factor of mDia1 is close to 1 (at low profilin concentrations), are the FH2 dimers assumed to nearly always be in a dissociative state? For formins with lower gating factors, how would the time spent in the "closed" FH2 conformation affecting the fitting for, or interpretation of, k_off_ rates?

5) It has been reported (Kovar et al., 2006) that the gating factor of mDia1 is profilin-dependent. Did the authors observe a decrease in the elongation rates mediated by the FH2-DAD construct in the presence of profilin? If so, this would suggest that the FH2 dimer begins to populate the non-dissociative, "closed" state for a more significant amount of time as the concentration of profilin increases. Would this affect the measured off-rates?

6) The authors interpret the stabilizing effect of profilin (in the absence of force) to be a result of the formation of a transient "ring complex" state in which profilin is bound both to the barbed end and to a polyproline track within the FH1 domain. Given the very weak affinities (>100 µM) of profilin for the barbed end and for short polyproline tracks such as those found in the FH1 domain of mDia1, what is the probability of populating this state, and how does it compare to the length of time spent in either of the dissociative states?

7) An alternative scenario that could account for the stabilizing effect of profilin on formin's processivity would be an allosteric effect of profilin binding to the FH1 domain. This profilin-FH1 complex would be populated to a higher degree than would the "ring complex" at low profilin concentrations. Was this scenario considered, and if so, why was it ruled out?

---

## [Author Response]

Essential revisions:There is quite a bit of variability in the specific rate constants between diverse formins. The impact of this investigation would certainly be enhanced if additional formins were assayed. The authors could study in particular the effect of profilin and force on the dissociation rate of another formin.

This is a good point, and we thank the reviewers for it. We chose mDia2, whose characteristics are quite different from that of mDia1. In particular, mDia2 is far more processive than mDia1. Nonetheless, we find that mDia2’s dissociation rate also decreases when profilin is added, and dramatically increases with tension (new Figures 3C, 3D and 4F). The text has been modified accordingly. In addition, since the modulation of mDia2-mediated elongation rate by force has never been measured, we now include it in new Figure 4—figure supplement 3. We believe these new results indeed enhance the impact of our investigation.

1) It would be extremely helpful for the clarity of the study to plot the survival fraction (of formin) as a function of actin subunits added, instead of as a function of time. This would immediately take into account differences in actin filament elongation rate that occur under the large array of conditions used in the study. This information is in the manuscript with regards to plots of dissociation rate as a function of elongation rate, but perhaps would make the data easier to understand. Furthermore, I believe plotting as a function of actin subunits added (instead of time) would allow for a better baseline with which to compare mDia1 (this study) and other studies that come out with other formins that have vastly different elongation rates over different conditions. This would also take into account changes in elongation rate caused by force (Figure 4).

We agree with the reviewers that providing information on the mean length of formin-induced filament elongation would be extremely helpful for the clarity of the study. In our Discussion, we have now inserted a new paragraph (subsection “Processivity mostly relies on FH2-filament interactions, with an unexpected contribution of FH1 domains”, last paragraph) and a new figure (Figure 5) that will hopefully give a more comprehensive picture of our findings to the readers: (1) profilin increases and (2) force dramatically decreases the formin-induced mean actin filament length. The graphs in this new figure also provide a direct comparison between mDia1 and mDia2.

However, in our other figures, we continue to plot survival fractions as a function of time, since this is how we derive the dissociation constants.

2) According to the Figure 1, actin filaments are around 30 µm. This is very long compared with the cellular context. How the length of the filament will affect the main conclusions presented here, especially on the tension effect?

The pulling force applied to formin scales with the filament length times the local flow rate. The filament length is one way to vary this force, but it has otherwise no impact on the formin properties (in Jégou et al., 2013, for example, we showed that the same elongation rate was obtained for different filament lengths, as long as the applied force was the same).

3) It would be helpful if the authors included a description of how they fit their survival plot data. I presume that they fit single-exponential decays to their data, but this is not explicitly stated in the text.

Indeed, in the absence of force, survival fractions were fitted by a single exponential function to extract formin dissociation rates. We have added additional explanations on our data analysis in the ‘Materials and methods’ section of the manuscript.

4) Since the gating factor of mDia1 is close to 1 (at low profilin concentrations), are the FH2 dimers assumed to nearly always be in a dissociative state? For formins with lower gating factors, how would the time spent in the "closed" FH2 conformation affecting the fitting for, or interpretation of, k_off_ rates?

First, we would like to stress that the gating factor of a formin does not provide a good estimate of the time it spends in the open state. As we have discussed in Jégou et al., 2013, the gating factor equals p_0_, the fraction of time spent in the open state in the absence of force, only if the on-rate for subunit addition in the open state equals the on-rate for subunit addition on formin-free barbed end, and there is no reason for this to be the case.

In our present model, we consider that the formin can dissociate from the open state and not from the closed state. The time spent in these two states is determined by the energy difference ϵ. A formin spending more time in the closed state would have a different value for ϵ and this would modify the numerical values obtained for the other terms, but not the overall aspect of the curves. As we write in the Discussion, “we focus less on obtaining precise fits to the experimental data – which are somewhat trivial and uninformative when a large number of adjustable parameters are involved – and instead demonstrate that the *qualitative* shape of the curves predicted by our model is consistent with our experimental measurements.”

5) It has been reported (Kovar et al., 2006) that the gating factor of mDia1 is profilin-dependent. Did the authors observe a decrease in the elongation rates mediated by the FH2-DAD construct in the presence of profilin? If so, this would suggest that the FH2 dimer begins to populate the non-dissociative, "closed" state for a more significant amount of time as the concentration of profilin increases. Would this affect the measured off-rates?

We also observed a decrease in the elongation rate mediated by mDia1(FH2-DAD) in the presence of profilin. This, however, does not necessarily indicate that profilin shifts the open-close equilibrium towards the closed state. For instance, profilin slows down the elongation of formin-free barbed ends, indicating that free profilin can slow down barbed end elongation by competing with monomers for the barbed end (Jégou et al., 2011, Pernier et al., 2016). Here, we show that, for a given elongation rate, the dissociation rate of mDia1(FH2-DAD) does not depend on the concentration of profilin (Figure 3E). This result indicates that, in the absence of FH1 domains, profilin simply reduces the dissociation rate by slowing down elongation, thereby reducing the number of transitions through the transient, dissociation-prone state that follows subunit addition. We now explain this point more clearly in the main text (subsection “Profilin increases formin processivity, involving FH1 domains”, last paragraph).

6) The authors interpret the stabilizing effect of profilin (in the absence of force) to be a result of the formation of a transient "ring complex" state in which profilin is bound both to the barbed end and to a polyproline track within the FH1 domain. Given the very weak affinities (>100 µM) of profilin for the barbed end and for short polyproline tracks such as those found in the FH1 domain of mDia1, what is the probability of populating this state, and how does it compare to the length of time spent in either of the dissociative states?

We have performed new experiments, whose results argue in favor of our interpretation that the formation of the ring complex is responsible for the profilin-induced reduction of formin dissociation (please see next point). We agree that the affinities of profilin for polyproline tracks and filament barbed ends seem to be weak. In that respect, the rapid acceleration of elongation with micromolars of profilin may appear as a paradox, indicating that these two affinities are not enough to describe the molecular mechanism. In this context, it is difficult to compute a probability for the formation of the ring complex. Nonetheless, we can assume that it is formed at least when delivering an elongation-producing profilin-actin subunit to the barbed end, and can thereby decrease the ensuing dissociation rate k_off_^T^. Since we propose that k_off_^T^ is the dominant contribution to dissociation at low force, this minimal frequency of ring complex formation would be enough to account for a significant decrease of dissociation rate. We now discuss this point in the main text (subsection “Processivity mostly relies on FH2-filament interactions, with an unexpected contribution of FH1 domains”, second paragraph).

7) An alternative scenario that could account for the stabilizing effect of profilin on formin's processivity would be an allosteric effect of profilin binding to the FH1 domain. This profilin-FH1 complex would be populated to a higher degree than would the "ring complex" at low profilin concentrations. Was this scenario considered, and if so, why was it ruled out?

We now test this alternative scenario explicitly, using profilin-R88E, a mutant that binds FH1 domains but not actin. We find that adding up to 20 µM of profilin-R88E does not reduce mDia1’s dissociation rate, indicating that profilin interactions with actin, including barbed ends, are required in order to enhance processivity. Simply loading the FH1 with profilin is not enough, and this result rules out the scenario of an allosteric effect. These new results are presented in new Figure 3—figure supplement 2 and in the main text (subsesction “Profilin increases formin processivity, involving FH1 domains”, last paragraph).